



# Beyond the stage-damage function: Estimating the economic damage on residential buildings from storm surges

Lea Skraep Svenningsen[1], Lisa Bay[1], Mads Lykke Doemgaard[1], Kirsten Halsnaes[1], Per Skougaard Kaspersen[1], and Morten Dahl Larsen[1]

[1]Climate Risks and Economics, DTU Management, Technical University of Denmark, 2800 Kongens Lyngby, Denmark

**Correspondence:** Lea Skræp Svenningsen (leasks@dtu.dk)

**Abstract.** Given the predicted global increase in extreme weather events, such as storm surges, the design of effective response strategies requires a very detailed and accurate understanding of the major factors driving damage costs. The costs of climate hazards are usually estimated using engineering approaches, which, based on different levels of building-specific information, link water inundation levels to the costs incurred by building owners. More recently, a number of scientific papers have pointed

to the limitations of such approaches because they omit important information about key context-specific factors such as emergency response options and a range of social factors reflecting age and social networks in the affected communities. This study contributes to this growing literature by providing rigorous and detailed econometric estimates of damage costs for residential buildings resulting from a storm surge that impacted large parts of Denmark in December 2013. We collected a comprehensive data set consisting of insurance cost data, the characteristics of individual buildings (size, age, construction

materials, heating source and distance from bodies of water), emergency services, previous experience with storm surges in the municipality and socio-economic factors. Our results indicate that the isolated effect of inundation depth on damage costs is highly sensitive to the inclusion of other explanatory variables. In our models the isolated effect of inundation depth is more than halved when our full set of control variables is included. Furthermore, our findings highlight the importance of controlling for spatial effects, such as the level of emergency services and socio-economic conditions. Discussing the transferability of our

findings, we highlight key sensitivities when using our damage functions in other contexts.

**Keywords:** Damage cost function, Storm surges, Climate adaptation, OLS regression, Flooding, Insurance data

## 1 Introduction

The damage costs of pluvial and fluvial flooding are usually estimated using engineering approaches, which typically focus on buildings and link water inundation levels to refurbishment costs. The simple approach using a relationship between inundation

depths and resulting damage costs alone (often referred to as the stage-damage function) forms the core basis of a vast number of studies and relates to direct damage alone (Smith, 1994; Middelmann-Fernandes, 2010). The damage cost functions in these models are often simple, and other key elements beyond building structure and refurbishment costs have not been considered, despite their importance in assessing damage cost levels. In this paper, we develop an extended cost model of building damage



from coastal flooding based on very detailed insurance damage cost data from a storm surge that occurred on 5 December

2013. This event caused severe damage in northwest Europe, including Denmark. Basing damage costs on insurance data
has its own merits and limitations. Insurance payments are revealed data and as such provide an accurate, often third-party
estimate of the damage costs, but they only capture one damage category, such as building and contents damage. However,
insurance schemes vary with respect to whether compensation covers the full price of refurbishing buildings, whether the age
of a building is reflected in the loss value and specific rules of self-payment if the same building has been repeatedly flooded.

Accordingly, insurance data on damage costs can be difficult to compare across different periods and regions. Insurance cost
data have been applied in multiple studies of both coastal flooding and pluvial and fluvial flooding (e.g. André et al. (2013);
Jonkman et al. (2008); Zhou et al. (2013)). Damage cost estimates can also be collected using survey methods. This approach
has the advantage that it can include more damage categories than are generally covered by insurance payments, but it suffers
from limitations such as personal reports being inaccurate due to limited memory or different understandings of what should

be included in specific cost categories. The choice of either method might also reflect aims, insured damage costs being of
interest to insurance companies, total damage costs to governments (local and national) and academics (Jongman et al., 2012).

To expand the simple stage-damage function, we combine the damage cost data based on insurance payments with house
characteristics, flood characteristics, socio-economic variables, the emergency response and experiences of flooding. Our con-
tribution to the literature is in line with several studies that have sought to refine the classic stage-damage function by introduc-

ing more influential parameters and/or variables into the calculations, but it concludes that their approach to estimating damage
cost functions still includes a lot of uncertainties. On the geophysical side, the additional factors often included in the literature
are flood duration, velocity, sediments and contamination, whereas the physical nature of houses (e.g. building materials) and
national or regional economies affect the variations in costs, and therefore geographical transferability, at a certain inundation
depth (Merz et al., 2010). The warning and forecast aspects and the experience and role of the community and one's neighbours

may also be important, as Grahn and Nyberg (2014)), for example, have pointed out. As a result of these additional influential
factors, the stage-damage function is characterized by a certain degree of scatter around the best-fit correlation. This has been
addressed as a severe challenge to studies of damage cost assessments by, for example, Smith (1994); Merz et al. (2014);
Prettenthaler et al. (2010); Moel et al. (2012); Prahl et al. (2018); Amadio et al. (2019).

Specifically, Carisi et al. (2018)highlight a significant increase in the predicted extent of flooding when a multivariate ap-

proach is used rather than using inundation depth alone, while at the same time highlighting the need for proper data. In
their study, building area and water velocity were found to be as important as maximum inundation depth. Along these lines,
Schröter et al. (2014) found that additional explanatory variables other than inundation depth improve the performance when
transferred to other settings both temporally and spatially. However, having used several damage-function models drawn from
the literature, they also emphasize that the overall structure of the model is a controlling factor in its performance.Jongman

et al. (2012) also employ several damage-function models taken from the literature and conclude that, other than the choice
of model, regional adjustments reflecting local economic conditions might be an important aspect requiring incorporation into
further and larger-scale studies. Ootegem et al. (2015) find improvements in predicting damage by introducing additional ex-
planatory variables, among them non-hazard indicators related to building characteristics, behavior and socio-economic factors.



These inclusions and the consideration of additional explanatory variables have also highlighted the need to consider intangible
variables such as adaptive capacity and preparedness (Merz et al., 2004), as important determinants of the damage costs.

Overall, the previous findings in the literature highlight the importance of including several explanatory variables when
estimating the economic damage costs from flooding, herein inundation depth, building construction materials, flood charac-
teristics and the adaptive capacity of the affected population. Furthermore, previous studies also highlight the sensitive nature of
damage cost estimates, discussing how transferable damage estimates are between storms, geography and over time (Cammerer
et al., 2013; Merz et al., 2014), highlighting the need to account for model uncertainty (Figueiredo et al., 2017) [1].

These studies are carried out especially within the fields of pluvial and fluvial flooding (e.g. Ootegem et al. (2015); Spekkers
et al. (2014); Merz et al. (2004, 2010); Amadio et al. (2019); Pistrika et al. (2014); Jongman et al. (2012)). ). In this paper, the
focus is on flooding from storm surges (coastal flooding), an area where the existing literature is scarce. Flooding from storm
surges has different characteristics from pluvial and fluvial flooding in that, for example, the water is saltwater, potentially
causing different types of damage. Transferring damage models estimated from pluvial flooding to coastal flooding might
therefore provide incorrect expected economic damage costs.

To study the importance of geography or the spatial effect on damage costs, in our dataset we have collected storm damage
costs from 29 different municipalities in Denmark that were affected by the same storm. This allows us to study how the same
storm can give rise to different damage costs across municipalities by controlling for differences in, for example, emergency
responses and socio-economic factors. Spatial effects have increasingly been recognized in both theoretical and applied econo-
metric work (Anselin and Arribas-Bel, 2013; Kelejian and Prucha, 2010), but also specifically in relation to damage costs from
flood events (Cammerer et al., 2013).

The novelty of this study lies in both its combination of different data sources and the variables it includes in the development
of a multivariable econometric model that estimates the expected damage costs from storm surge-induced coastal flooding.
By coupling influencing variables from a comprehensive set of sources and sectors, including new insurance data, national
emergency management data, flood simulations and building characteristics, we are able to control rigorously for a great
number of parameters. In the literature all these have previously been identified separately as important determinants of damage
costs but, to the best of our knowledge, have never been analyzed within the same econometric framework.

## 2 Materials and methods

To perform our econometric estimate of damage costs, we rely on several different sources of data, which are presented in the
subsections below. The specification of the econometric models is given in Section 2.2.

### 2.1 Data

In this study, we combine variables from six unique datasets. Table 1 presents each of the included variables, their units and
levels, and their dataset source. The following sections present each dataset individually.

---

[1]Some recent papers have investigated and compared the performance of individual models and model ensembles, see Figueiredo et al. (2017, 2018)



**Table 1.** Description of variables in the final data set

| Variable | Unit and range | Source | Spatial level |
|---|---|---|---|
| **Building damage** | | | |
| Damage cost | €: 0.18-497.12 thousand Euro | Danish Storm Council | Building |
| **Storm surge characteristics** | | | |
| Return Period | Years: 7-1000 yr | Flood modelling; Flood water statistics | Building |
| Depth | Cm: 1-296cm | Flood modelling | Building |
| **Building characteristics** | | | |
| Size | $m^2$: 45-364 $m^2$ | BBR | Building |
| Rooms | Number: 2-9 rooms | BBR | Building |
| Age | Years in 2016: 0-239yr | BBR | Building |
| Floor | Number (only groundfloor incl.) | BBR | Building |
| Building type | Dummy according to type | BBR | Building |
| Heating source | Dummy according to type | BBR | Building |
| Outer construction | Dummy according to type | BBR | Building |
| Renovations | Years since last: 0-91 years | BBR | Building |
| Distance to lake | Meter: 22.5-1566.2m | Tokes data(?) | Building |
| Distance to coast | Meter: 3.4-2817.7m | Tokes data(?) | Building |
| Distance to habour | Meter: 3.4-7997.0m | Tokes data(?) | Building |
| **Flood experience** | | | |
| Previous storms | Number of insurance claims: 0-602 | Danish Storm Council | Municipality |
| Previous storms cost | Index 100= highest expenditures | Danish Storm Council | Municipality |
| **Emergency Management Agency Services, EMAS** | | | |
| EMAS duration | Hours: 0-582hr | Danish EMAS | Municipality |
| EMAS count | Number of services: 0-38 | Danish EMAS | Municipality |
| **Social vulnerability indicators** | | | |
| Retired persons | Percentage: 19-35% | Statistics Denmark | Municipality |
| Income per capita | €: 37-80 thousand Euro | Statistics Denmark | Municipality |
| Expenditure on medical consultantions | €: 292-404 Euro/capita | Statistics Denmark | Municipality |

Note: The reference year for the damage cost from the Danish Storm Council is in current prices through the period 2013-2017. The reference year for income per capita and expenditure on medical consultations is 2019. Reference categories for dummy variables in subsequent econometric models are: Heating source = district heating and Outer construction = wood




### 2.1.1  Building damage

From the Danish Storm Council (DSC), we obtain realized insurance claims at the household level. The DSC (DSC, 2019), is an independent council that falls under the Danish Ministry for Business and Growth. The DSC decides whether an official storm surge has occurred, since insurance claims can only be made under these circumstances. The DSC then handles all insurance claims and compensation related to the event. Compensation is only granted if the water level has a recurrence interval of at least twenty years. The DSC covers the majority of all damage to building structure and chattels, with a few exceptions such as terraces, separate garages and carports, and moveable property stored in basements and rooms below ground that are not approved for living [2]. The DSC also grants compensation for the costs of cleaning and rehousing. The data cover the period since 2013, when all claims were collected at the address level and were divided into residential, leisure (summer cottages) and business. The dataset does not contain any information on the inundation depth causing the damage. The storm surge of 2013 resulted in nation-wide economic losses, with insurance payouts for residential building owners in 29 out of 97 Danish municipalities. 2,761 individual insurance claims were made, of which 2,136 received payouts from the DSC, amounting to a total of approximately 130m €. In developing the econometric model we only use data for building damage to residential buildings, for which we have 551 unique observations covering a third (45m €) of the total insurance payouts (1). The full descriptive statistics from the DSC are available listed from in Table A1 in the Appendix.

### 2.1.2  Storm surge characteristics

One of the main factors influencing the shape and size of the damage cost function is the inundation depth (Amadio et al., 2019). Because the DSC data do not contain this information on a household basis, inundation depths are estimated by simulating flooding during the storm surge using a simple static 2D surface flood model framework. A high-resolution digital elevation model (DEM) is combined with data from a network of water-level measuring stations indicating the maximum water level at sea and the corresponding return period during the storm surge (DCA, 2018). The outputs of the flood model are flood hazard maps showing the maximum extent and depth of flooding for the different geographical areas. To obtain inundation depths, the nearest measuring station for each of the DSC-registered insurance claims was found, and flood simulations were conducted, raising the sea level to match the observed maximum water level for each observation. Only areas that are directly connected to the sea are considered as having been flooded during the storm surge. The DEM has a pixel size of 0.4 m but is aggregated to a 10 m resolution for operational simulation times. Furthermore, all protective infrastructure such as floodgates and sluices are closed during the flood simulations, as would be the case during a storm surge. To quantify the performance of our flood-model simulation, we evaluated accuracies for stand-alone residential buildings by spatially comparing the DSC data (insurance claims) with the flood hazard maps. We selected a hot-spot area near Roskilde Fjord for evaluation, as this represents a large share of the insurance payouts, see Figure 1. A three-step procedure was used for the evaluation. First, we assigned each of the stand-alone buildings to the closest water-level measuring station (three stations are located in this area).

---

[2]For a complete list of losses not covered by the DSC (in Danish): Stormrådets Dækningsvejledning, 28. juni 2018. A simplified list in English: https://www.danishstormcouncil.dk/artikler/danishstormcouncil/storm-surge/how-to-receive-compensation-after-a-storm-surge/

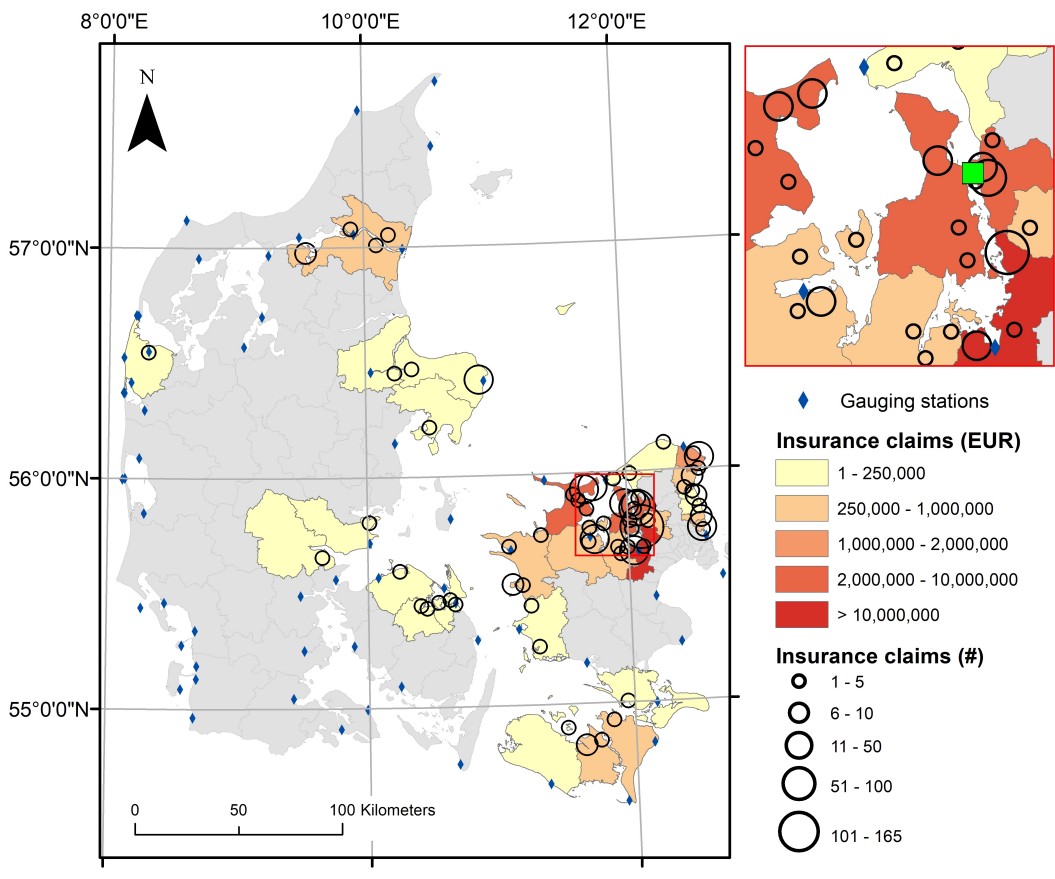

**Figure 1.** Insurance claims for residential buildings in Denmark following the storm surge of 5 December 2013. The green square in the top right section marks the hot-spot, analysed in Figure 2. The GIS layer for the Danish administrative regions is open-source and obtained through: https://sdfe.dk/hent-data/danmarks-administrative-geografiske-inddeling, ©SDFE

Secondly, we conducted three individual flood simulations, one for each subset of buildings linked to a specific measuring station. Finally, we evaluated the number of accurately and inaccurately flooded buildings by means of an overlay analysis of the flood-hazard maps and the DSC data, assuming that the latter represent the "ground truth".

Figure 2 gives the results of the model evaluation when simulating storm surge "Bodil" for Frederikssund municipality in the eastern part of Roskilde Fjord, Denmark, which suffered large-scale economic damage during the storm surge. The flood model captures 67 of the 70 buildings with DSC insurance claims, but over-estimates the total flooded area, as 38 buildings without insurance claims are flooded in the model. However, we do not know whether these buildings actually experienced high water levels during the storm surge, but were better protected and therefore did not experience any economic losses. However, a general concern in using static flood models is that the flooded area may be over-estimated, as the model does not consider the

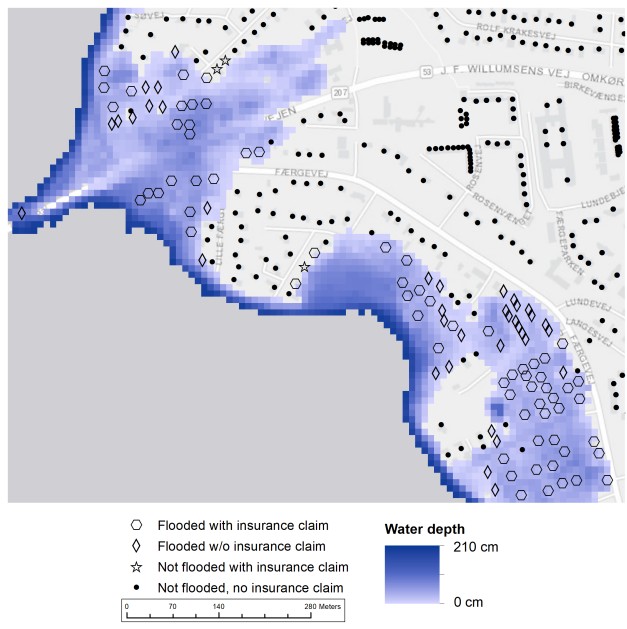

**Figure 2.** Results of flood model evaluation for Frederikssund municipality. The GIS layer for the Danish coastal line is open-source and obtained through: https://sdfe.dk/hent-data/danmarks-administrative-geografiske-inddeling, ©SDFE

duration of the storm surge and thus envisages low-lying, inland areas being flooded that the water might not reach. In addition, the model does not account for the natural infiltration and filling of drainage systems, which contribute to the over-estimates.

### 2.1.3 Buildling characteristics

The Danish Building and Housing Register (BBR) contains information on building and housing characteristics for all developed properties in Denmark. The register contains a large variety of variables for each building, such as type, year of construction, number of rooms, size, garage and garden size, as well as technical details like building materials and heating type (BBR, 2019).

### 2.1.4 Flood experience

As an indicator of the experience of flooding, we use insurance data from the DSC for previous storm surges from 1999-2008 aggregated at the municipality level. As shown in Table 1, the two indicators are i) the number of insurance claims and ii) an index number of the total compensation sum with index 100 corresponding to approximately 10,700,000€, which corresponds to the highest claim within a municipality in that period[3]. In order to capture the relative differences in the previous size of

---

[3]The compensation sum cannot be compared with the 2013 sum, as the legislation was changed due the scale of the damage from the storm surge, resulting in a vast increase in the compensation.



insurance claims between municipalities, we use an index instead of the actual insurance claim. In estimating our model to account for multicollinearity, only the number of insurance claims is used.

### 2.1.5 Emergency Management Agency Services, EMAS

AA dataset consisting of all emergency and rescue services involved in the storm surge for the period 4-12 December 2013 was retrieved with the permission of the Danish Emergency Management Agency (DEMA, 2019). The dataset covers emergency and rescue services relating to both strong winds, high water levels and inundation. To isolate the effect of the emergency services for storm-surge issues, data related to strong wind damage was removed (e.g. roofs blown off, trees falling on roads inland). The services include pumping out water, installing water tubes, providing sandbags and personal rescues. For every

action, the data provide information on the time spent on the service and the number of persons involved in it. However, the data on the number of staff involved in a given action are estimates based on the time spent and the number of cars sent. Due to uncertainties in these estimates, we only include time spent on a given action in the dataset. The data are on the address level. However, since the emergency services' activities will often benefit larger areas, the data points have been summarized for each Danish municipality. Data on emergency and rescue services therefore relate to the municipal level.

### 155 2.1.6 Social vulnerability indicators

Socio-economic data were retrieved from the Statistics Denmark database, StatBank (StatisticsDenmark, 2019). Data on income per capita, public expenditure on medical consultations and retired population in 2013 were retrieved for all 98 municipalities in Denmark. To enable comparison between municipalities, both public expenditure on medical consultations and size of retired population were normalized in accordance with the total population size of each municipality. Data on population

size in 2013 were therefore also retrieved from Statistics Denmark (StatisticsDenmark, 2019).

### 2.1.7 Construction of dataset

The final dataset is cross-sectional, consisting of 551 observations (unique residential buildings) in relation to the storm surge of 5 December 2013, with a geographical coverage of 29 Danish municipalities. In the study, observations of residential buildings that are listed as an apartment on the first floor and above, as well as observations of no building insurance being claimed but

only inventory insurance from the storm surge being collected, were omitted.

### 2.2 Econometric specification

### 2.2.1 Multiple regression analysis

Within economics, multiple regression analysis using Ordinary Least Squares (OLS) is a widely used econometric method of constructing multi-variable models from empirical data (Wooldridge, 2013). The method has been applied in a few studies on

the determinants of damage costs for flooding events on building (Ootegem et al., 2015), but in some cases the analyses lack sufficient data for the method to be firmly established (Komolafe et al., 2019; Romali et al., 2019). This indicates a need for




additional studies using econometric methods to estimate the damage from flooding events. The general model is expressed as Equation 1.

$$Damagecost = \beta_0 + \beta_1 X_1 + \beta_2 X_2 + ... + \beta_k X_k + \epsilon \tag{1}$$

where each of the slope coefficients is a partial derivative of *y* with respect to the *x* variable it multiplies. That is, holding all other *x*'s fixed, $\beta_1 = \partial y / \partial x_1$ (Wooldridge, 2013). Efficient and non-biased estimates for all parameters can be obtained as long as the basic OLS assumptions are respected. Ensuring linearity in the relationship between the dependent and the independent variables, the model can be estimated as a log-log or log-linear model in order to assess the relative impact of variance in the independent variables on the dependent variables.

### 180   2.2.2   Variable selection and functional form

For each observation, the original dataset included more than eighty variables, with detailed variables for building characteristics, location, storm characteristics, previous storms, insurance claims and the level of emergency services provided. The literature review provided input into the initial selection of variables, identifying those that several studies have found to be significant predictors of the economic damage from a flooding event (Merz et al., 2013; Zhai et al., 2005; Elmer et al., 2010).
In this paper, we apply a two-stage variable selection and definition of functional form in our econometric models. In the first step, we used the R software package 'PanJen' (Jensen and Panduro, 2018) to identify the functional form for each potential explanatory variable. A visual investigation of the relationship between the explanatory variables and the dependent variables indicated that most or some of the relationships cannot be characterized as linear. As a result, we applied the semi-parametric tools in PanJen to compare the performance of different functional forms of the explanatory variables.
To detect the additional predictive power of different types of data, the regression analysis was performed by systematically adding explanatory variables and assessing the resulting statistic, yielding a total of three different models. An overview of the variables included in the final regression models is given in Table B1, while an overview of our a priori variable hypotheses can be found in Table E1, both found in the Appendix. The initial model (Model 1) was estimated based on flooding depth alone. For Model 2, fourteen variables of housing characteristics were added (size, age, heating source, construction materials
and distance from bodies of water). Model 3 also included eight variables reflecting the spatial variation in damage costs by including the return period of the storm surge, previous storm surge experience and emergency services in the municipality, in





addition to variables indicating social vulnerability at the municipal level (retired persons, income and public expenditure on medical consultations). The regression equation for Model 3 is given in Equation 2.

$$
\begin{aligned}
\text{Damage}_i = {} & \beta_0 + \beta_1 ln(depth_i) + \\
& + \beta_2 ln(size_i) + \beta_3 age_i + \beta_4 renovated_i \\
& + \beta_5 centralheating_i + \beta_6 stoveheating_i + \beta_7 electricheating_i + \beta_8 heatpumpheating_i \\
& + \beta_9 lightweightconcrete_i + \beta_{10} timbered_i + \beta_{11} brick_i + \beta_{12} concrete_i \\
& + \beta_{13} ln(coastdist_i) + \beta_{14} ln(lakedist_i) + \beta_{15} ln(habourdist_i) + \\
& + \beta_{16} ln(RP_i) + \beta_{17} prevcount_i + \beta_{18} EMAS\_duration_i + \\
& + \beta_{19} ln(depth)_i * EMAs\_duration_i + \beta_{20} ln(rp)_i * EMAS\_duration_i \\
& + \beta_{21} retired_i + \beta_{22} income_i + \beta_{23} med\_exp_i + \epsilon_i
\end{aligned}
\tag{2}
$$

## 3 Results and discussion

**OLS regression results**

All regression models were estimated using Stata 13 (StataCorp, 2013) ], and the results are presented in Table 2. All mentions of statistical significance in what follows refer to a 5% significance level, although Table 2 also reports significance levels of 1% and 10%. The initial and simple model (Model 1), based on inundation depth alone, was included in the study to depict a baseline model in order to identify the change in predictive power when more explanatory variables are added, as in Models 2 and 3. Across all three models, our results indicate that an increase in water depth significantly increases damage costs, which confirms both intuition and previous findings in the literature (Messner, 2007).Furthermore, our results also indicate that individual building characteristics influence damage costs: for example, larger buildings have significantly greater damage (Model 3), and buildings heated by electric and central heating have significantly larger damage costs (Models 2 and 3). In addition, adding spatial variables, such as municipal characteristics, increases explanatory value (higher adjusted R2) without changing the significance levels of most of the variables. Model 3, which captures most of the spatial effects, shows especially the importance of including municipality characteristics, including social vulnerability, resources, the emergency response, etc.

Our preferred model, Model 3, is an extended model including both building characteristics and variables in order to capture specific spatial variations. We include a continuous measure for the return period of the storm surge, *rp*, and a variable, *prevcount*, that measures historical exposure to storm surges at the municipal level, as captured by the number of historical insurance claims. In addition, we include variables related to the national Emergency Management Agency's Services (EMAS) during the storm surge, which controls for the duration of the effort (*EMAS_duration*). This variable is linked to the inundation depth and return period, since it is expected that the emergency services have greater representation in areas with higher inundation depths. Furthermore, three spatial variables regarding the socioeconomic characteristics of the municipalities were




**Table 2.** OLS regression results

|  | Model 1 | Model 2 | Model 3 |
|---|---|---|---|
| ln(depth) | 0.572*** | 0.474*** | 0.375** |
|  | (0.046) | (0.044) | (0.079) |
| ln(size) |  | 0.263 | 0.464*** |
|  |  | (0.163) | (0.169) |
| age |  | -0.007*** | -0.004** |
|  |  | (0.002) | (0.002) |
| central_heating |  | 0.342** | 0.273** |
|  |  | (0.168) | (0.165) |
| stove_heating |  | 0.416 | 0.496 |
|  |  | (0.372) | (0.461) |
| electric_heating |  | 0.730*** | 0.559*** |
|  |  | (0.211) | (0.192) |
| heatpump_heating |  | 0.545** | 0.332 |
|  |  | (0.224) | (0.211) |
| ln(rp) |  |  | 0.121 |
|  |  |  | (0.079) |
| PrevCount |  |  | 0.001 |
|  |  |  | (0.001) |
| EMAS_duration |  |  | -0.013** |
|  |  |  | (0.005) |
| ln(depth)*EMAS_duration |  |  | 0.000 |
|  |  |  | (0.000) |
| ln(rp)*EMAS_duration |  |  | 0.002** |
|  |  |  | (0.001) |
| Share retired persons |  |  | 2.681 |
|  |  |  | (2.481) |
| income per capita |  |  | -0.000 |
|  |  |  | (0.000) |
| medical expenses |  |  | -0.001*** |
|  |  |  | (0.001) |
| Constant | 10.593*** | 9.571*** | 11.865*** |
|  | (0.165) | (0.920) | (1.412) |
| Construction variables |  | yes | yes |
| Distance to water bodies |  | yes | yes |
| Observations | 551 | 551 | 551 |
| Adjusted $R^2$ | 0.211 | 0.321 | 0.367 |

Robust standard errors in parentheses

* $p < 0.10$, ** $p < 0.05$, *** $p < 0.01$

EMAS = Emergency Management Agency Services

added. As described earlier, these variables are included to capture any difference in damage costs at the spatial level of the municipality with regard to how vulnerable local communities are. The hypothesis is that the vulnerability of a local community could influence the municipality's overall efforts to reduce the expected damage costs so that municipalities that are more vulnerable have less economic and social capital with which to respond to extreme events such as storm surges. The overall explanatory power of the model as captured by the adjusted R2 is 0.37. The increase in the explanatory power from





0.21 in Model 1 and 0.32 in Model 2 reflects the finding of Ootegem et al. (2015), who also saw an improvement by including variables for socio-economic status and building characteristics.

Model 3 shows a statistically significant increase in the resulting damage of 0.38% per 1% increase in inundation depth, a positive relationship also identified previously in the literature (Jonkman et al., 2008). The effect on damage from the inundation depth is smaller compared to Models 1 and 2, indicating the presence of omitted variable bias in those models.

We find that several of the building characteristics, such as size, age and heating source, significantly impact on the size of the damage costs. Larger buildings have statistically significant greater damage, with a 1% increase in the size of the building leading to an increase of 0.46%. The finding of a positive relationship between size and damage cost is confirmed by a study by Carisi et al. (2018), who also find a significant, positive relationship.Older buildings are subject to lower expected damage costs, with a ten-year increase in a building's age reducing damage by 4%. Interpreting the causal effects of age upon damage costs is

not straightforward. It is likely that the observed negative effects of age on damage costs could be due to older buildings having been built in safer locations, for example, in areas not prone to flooding. There has been an increasing tendency in Denmark, as well as globally, for the development of new settlements and neighborhoods to be located closer to flood-prone areas (Seto et al., 2011; Small and Nicholls, 2003), presumably motivated by the amenity benefits gained from proximity to the water. This trend could explain why older residential buildings are expected to have lower damage costs than newer buildings.

Furthermore, our results show that, compared to buildings heated by district heating, those heated by an electrical heating system suffer more damage, with a figure of 63.15% as opposed to 47.67% [4].Of the 551 observations in our data, 241 buildings have central heating, 139 have electric heating, 64 buildings are heated by a heat pump and five have stove heating (see Table **??** in the Appendix). The reference category, district heating, is only observed for 102 buildings. Nevertheless this category was chosen as a reference since the four remaining categories are all characterized by being building-specific and thus are

potentially more susceptible to damage than a district heating system would be. So, although these percentage effects seems large compared to the other effects we found, they essentially capture the greater sensitivity of a localized heating system compared to a network-based system and highlight the sensitivity of these systems to a storm-surge event. The interpretation of the heating systems' impact on damage costs could reflect the locations of different heating systems within buildings and thus the costs of repairing damaged heating systems. Confirming this hypothesis, heat pumps and electric heaters, which we

find contribute most to damage costs, are often placed in low-to-ground positions within buildings.

The direction and significance of both the age of a building and whether it has electrical and central heating remains unchanged between Models 2 and 3, suggesting that the relevance of these variables is robust to the inclusion of the specified spatial variables. However, the size of the effect is reduced, and the heat pump does not seem to suffer greater damage than the district heating systems in Model 3. Nonetheless the effect of size is only significant in Model 3, which indicates that

including important spatial variables reduces an omitted variable bias. We find no indication in any of the models that exterior construction materials influence damage costs, nor distance from the nearest coast or lake. Also, the results indicate that the storm intensity ($rp$) and the number and severity of prior storm surges have no significant effect on the damage costs.

---

[4]For the dummy variables capturing heating source, their effect must be specifically calculated and cannot be directly read from Table 2. The formula we used for calculating the effect of dummy variables is: $100*(\exp(\beta_x - 0.5\beta^2_{sderror}) - 1)(Kennedy\,et\,al.,\,1981)$





Interestingly, the variables regarding emergency responses show a significant decrease in damage costs the more time is spent on emergency actions. There is a larger decrease in damage costs at lower inundation depths, as captured by the partial effect of the interaction term (*EMAS_duration + ln(depth)\*EMAS_duration*). The calculated reduction in damage per extra hour spent on emergency management ranges from 1.16% for the lowest recorded inundation depth to a reduction in damage of 0.16% for the highest recorded inundation depth.

We find no effect of either the share of retired persons in a municipality or the average income per capita on damage costs. However, the results indicate that municipalities with greater public expenditure on medical consultations also have statistically lower damage costs. Intuitively, we would have assumed that municipalities with higher public expenditure on medical consultations indicate a vulnerable community, which would suggest that the effect on damage costs should be positive. On the other hand, this could also be seen as an indication of a resourceful community in which people act on their health problems, which could explain the negative relationship we observe.

To sum up, our results indicate a substantial increase in explanatory power from 21% to 39%, going from a simple linear regression including only inundation depth (Model 1) to a multivariable regression model (Model 3). Furthermore, our study shows that adding explanatory variables besides inundation depth decreases the effect of inundation depth alone, for example, a fall from a 0.57% increase in damage costs to 0.37% from a 1% increase in inundation depth – a substantial difference. Thus, our study indicates that using only the simulated inundation depth to predict damage costs could lead to these costs being over-estimated. Specifically, the inclusion of spatial variables changes the effect size of several of the explanatory variables, adding to the explanatory power of our model.

If OLS regressions are to provide unbiased and consistent estimates, we have to assume that observations are independent of one another (Lesage, 2014). In this case, we might suspect the presence of neighborhood effects that influence damage costs at a lower level than the municipal level. As pointed out by Anselin and Arribas-Bel (2013), it is only in very restricted cases that fixed spatial effects, such as those we include in Model 3, correctly account for the possible spatial dependence between observations. An example of such a small-scale spatial effect could be the degree of social cohesion among neighbors, which potentially could influence their ability or willingness to help each other during an extreme event such as a storm surge. Not controlling for such a potential effect in the model set up means that spatial auto-correlation might be present, leading to biased and inconsistent OLS estimators. To investigate whether Models 2 or 3 display any spatial auto-correlation, we compare model performance based on Moran's I (see Table F1 in Appendix), indicating that spatial autocorrelation is present in Model 2 (Moran's I= 0.117\*\*\*). However, in Model 3, which includes spatial variables at the municipality level, we find that the evidence of significant spatial autocorrelation has decreased (Moran's I = 0.062\*\*). In particular, the test for spatial autocorrelation in the error term is only significant at a 10 % level, suggesting that the model accounts for most of the unobserved spatial effects. This again highlights the importance of including variables representing spatial variance and not just focusing on specific house or individual characteristics. The indication that some spatial dependence is still present in the extended model suggests that future studies could explore this further using, for example, spatial econometric methods (LeSage and Pace, 2009; Cliff, 1973).





## 4 Conclusion

Multivariable damage cost models of storm-surge flooding are essential in supporting cost-effective investments in adaptation options. First, they can highlight individual or combinations of variables that are driving high economic losses, showing which
variables are particularly important to address in devising risk-reduction strategies. Secondly, the models can be used to prioritize actions between different geographical areas and provide an estimate of how much society should be willing to invest in adaptive measures. Our results confirm that multivariable models can provide more accurate results than a simple model. Through the inclusion of a wide set of explanatory variables in the damage costs of coastal flooding, our study particularly highlights the necessity of controlling for spatial variation in estimating damage costs, whether in relation to socio-economic
conditions, flooding characteristics or emergency services. Including more variables increases the explanatory power of our model from 21% in the simple model, where inundation depth is the only variable, to 39% in the multivariable model. Several different variables are found to significantly influence damage costs. Besides inundation depth, they include building characteristics (size, age), emergency response efforts and type of heating source. In particular, the presence of electric heating systems appears to be surprisingly sensitive to flooding, as damage costs are found to be much higher for buildings with such systems
compared with, for example, houses drawing on district heating. The chief benefit of our approach is the stringent econometric method we have used. In theory, this approach should facilitate transferability to other settings similar to those in our study. We are aware that the type of data used for this model is not widely accessible in all regions and countries, which is why the results of Model 3 will not easily be transferred to other regions. However, our study still highlights the importance of including variables that can account for differences across municipalities and regions. The results of our study should thus be replicated
in other settings before being used as input to adaptation option strategies.





**Table A1.** DSC Insurance data for the storm surge of 5'th December 2013

| | Residential | Share | Leisure | Share | Business | Share | Total |
|---|---|---|---|---|---|---|---|
| **Insurance claims (#)** | 1294 | 47% | 1098 | 40% | 369 | 13% | 2.761 |
| **Insurance payouts (#)** | 927 | 43% | 942 | 44% | 267 | 13% | 2.136 |
| **Payouts () €** | 61,195,309 | 47% | 50,301,761 | 39% | 17,726,855 | 14% | 129.223.925 |
| | **Payout, €** | **Share** | | | | | |
| **Building** | 106,778,147 | 83% | | | | | |
| **Chattel** | 12,534,970 | 10% | | | | | |
| **Rehousing** | 2,434,311 | 2% | | | | | |
| **Own risk** | 7,476,496 | 6% | | | | | |
| **Total** | **129,223,925** | | | | | | |



**Table B1.** Variables included in final regression models

| Variable | Model 1 | Model 2 | Model 3 |
|---|---|---|---|
| **Building damage** | | | |
| Damage cost | X | X | X |
| **Storm surge characteristics** | | | |
| Inundation depth | X | X | X |
| Return Period | | | X |
| **Building characteristics** | | | |
| Size | | X | X |
| Age | | X | X |
| Heating source (dummy: central, stove, electric, heat pump, district) | | X | X |
| Outer construction (dummy: wood, timbered, concrete, lightweigt concrete, bricks) | | X | X |
| Renovations (dummy: yes, no) | | X | X |
| Distance to lake | | X | X |
| Distance to coast | | X | X |
| Distance to habour | | X | X |
| **Flood experience** | | | |
| Number of previous storms | | | X |
| **Emergency Management Agency Services, EMAS** | | | |
| EMAS duration (hours spent) | | | X |
| **Social vulnerability indicators** | | | |
| Population share of retired persons | | | X |
| Income per capita | | | X |
| public expenditures on medical consultations | | | X |

Reference categories for dummy variables in subsequent econometric models are: Heating source = district heating and Outer construction = wood





**Table C1.** Descriptive statistics for continous variables

|  | mean | sd | min | max |
|---|---|---|---|---|
| damage costs (y) | 501441.02 | 564825.10 | 1455.00 | 3728359.25 |
| depth | 44.58 | 53.01 | 1.00 | 296.00 |
| rp | 687.21 | 393.57 | 7.00 | 1000.00 |
| size | 145.61 | 52.80 | 45.00 | 364.00 |
| age | 61.05 | 44.95 | 0.00 | 239.00 |
| ln(coast_dist) | 4.66 | 1.18 | 1.24 | 7.94 |
| ln(lake_dist) | 5.96 | 0.77 | 3.12 | 7.36 |
| PrevCount | 39.19 | 91.14 | 0.00 | 602.00 |
| PrevCostIndex | 4.29 | 12.95 | 0.00 | 100.00 |
| EMAS_duration | 262.72 | 213.25 | 0.00 | 582.40 |
| EMAS_count | 19.98 | 14.71 | 0.00 | 38.00 |
| ln(depth)*EMAS_count | 71.57 | 59.40 | 0.00 | 173.84 |
| ln(rp)*EMAS_count | 133.95 | 105.52 | 0.00 | 262.49 |
| ln(depth)*EMAS_duration | 955.98 | 888.76 | 0.00 | 3314.08 |
| ln(rp)*EMAS_duration | 1770.74 | 1512.74 | 0.00 | 4023.10 |
| retired persons | 0.23 | 0.04 | 0.19 | 0.35 |
| income per capita | 376151.35 | 63298.34 | 280394.00 | 599126.00 |
| medical expenses | 2550.63 | 173.29 | 2194.22 | 3031.16 |



**Table D1.** Descriptive statistics for dummy variables

| Construction variables | Observations |
|---|---|
| renovated | 202 |
| lightweight_concrete | 33 |
| timbered | 17 |
| brick | 362 |
| wood (reference) | 131 |
| central_heating | 241 |
| stove_heating | 5 |
| electric_heating | 139 |
| heatpump_heating | 64 |
| district_heating (reference) | 102 |



**Table E1. Variable specific hypotheses.**

| Variable | Sign | Description |
|---|---|---|
| depth (cm) | + | A higher water depth will increase expected damage |
| return period (years) | + | A higher return period equals harsher and more seldom storms, which equals higher damages |
| age (years) | +/- | Older house may be more vulnerable, but may on the other hand also be built in more safe areas |
| size (m2) | + | A larger house will, ceteris paribus, incur more damages |
| renovated (dummy) | + | The longer time it has been since a major renovation, the less prepared the property is for storm surge |
| exterior material (wood, timbered, brick, lightweight concrete, concrete) | +/- | The type of exterior materials is likely to both increase (wood, timbered) or decrease (concrete, lightweight-concrete, brick) expected damage costs |
| heating system (heat pumps, stoves, central, and district heating) | +/- | The source of heating might influence the total damage amount, depending on how sensitive the system is towards flooding. Stoves, heat pumps and central heating systems could be more vulnerable to flooding compared to district heating |
| distance to coastline (meter) | - | The longer away a property is from the coastline, the lower the damages it incurs |
| previous insurance claims in area due to storm surge (count) | - | If buildings in the municipality has been flooded before due to prior storm surges, they might be more prepared for the next one and thus have lower damage costs |
| previous insurance relief payment in area due to storm surge (index) | - | If buildings in the municipality earlier have had high insurance payments due to storm surges, we could expect building owners to be more prepared for following storms. |
| Time spent in a given municipality on flood control from the Emergency Management Agency (hours) | - | This variable has to interacted with the RP and depth of the storm surge to account for more time spent in areas with more severe storm surges. When this is done, more time spent on emergency response, should result in lower damages on buildings. |



**Table F1. Test of global spatial autocorrelation in OLS residuals**

| Model | Moran's I | Robust LM-test for spatial error |
|-------|-----------|----------------------------------|
| Model 2 | 0.117*** | 10.42*** |
| Model 3 | 0.062** | 2.86* |

$^{*}\ p < 0.10,\ ^{**}\ p < 0.05,\ ^{***}\ p < 0.01$



*Author contributions.* L.S.S., L.B. and M.L.D. designed the analysis, L.S.S. and L.B ran the analysis. All authors contributed to the draft of the paper.

*Competing interests.* No competing interests are present

*Acknowledgements.* The authors would like to thank the DSC and EMAS for kindly making their data available and sharing it for the analysis.
We would also like to acknowledge the support of the Innovation Fund Denmark, funding the COHERENT research project (7048-00004B).



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
