# Peer review of "Beyond the stage-damage function: Estimating the economic damage on residential buildings from storm surges"

_Natural Hazards and Earth System Sciences, 2020_

## Referee Comment (RC1) · Anonymous Referee #1 · 25 Mar 2020

**Brief summary**

The authors investigate the economic flood damage to private households caused by a storm surge event in Denmark based on an extensive data set provided by various danish official institutions. The data set contains a broad range of variables including building damage, flood experience, emergency measures and social vulnerability indicators. The characteristics of the storm surge are estimated by flood models. The authors develop three regression models based on this data set using three different variable sets. Based on the explanatory power of these models the authors conclude that the influence of the inundation depth is more than halved when using other ex-

planatory variables additionally. Another finding is that multi-variable models give more accurate results than simple models. The authors also found variables representing social conditions or level of emergency measures in an area to be important.

The topic of the manuscript is interesting and certainly fits to the Journal's scope. From my point of view, the main novelty of the manuscript is the inclusion of national emergency actions in the models. The manuscript is mostly clearly written. However, it could benefit from a better structure of text body (e.g. by the inclusion of more paragraphs). Some methods and motivations need to be addressed more clearly. At the moment the manuscript lacks some additional figures (e.g. correlation matrices, distribution of the variables, maps of the residuals). All results which are essential for the main points of the study should be presented within the main text. Some of the conclusions are currently not sufficiently supported by the results. A more detailed validation of models and some adaptations (e.g. the use of the relative damage in the models) is required and could help to support the main points or even facilitate to draw some additional conclusions.

[Figure]

**Broad comments**

- In general the readability of the text would benefit from the use of more paragraphs and better structuring of the text body. Numbers between one and twelve should be written as words, all greater than twelve as a number (e.g. in lines 95 and 181).

- The introduction could use a review on recent advancements on machine learning methods used in estimated flood damage. Also the question why you used a different approach should be dealt with as you mention the choice of your approach as a benefit in the conclusion. This could potentially strengthen your point.

- I guess most of the identified variables have been included in other damage models before (also in papers you already cite as e.g. Merz et al. 2013 or Schröter et al. 2014). To my knowledge the inclusion of national emergency actions has not been done before and is indeed a novel point.

- Usually the relative damage (absolute damage divided by absolute asset values) is used to make different buildings comparable with each other. The use of the absolute damage in your case could also explain that age and size have a significant influence on the damage costs.

- Correlation matrices could help to get an idea of the multicollinearity.

- I think the distributions of the used variables could be presented a bit more detailed (e.g. by the means of violin plots).

- Why no detailed cross-validation? You could even make validation with spatial transfer of the models between the different municipalities.

- Morans I should be explained in the method section. You could also write a bit more about the motivation as not every reader might be familiar with this. In addition, you could also plot the residuals spatially to check for patterns and also as a histogram to check whether they are normally distributed or not. If not the morans I might have a limited meaning due to a "biased" mean.

- Results (such as on the autocorrelation) which are essential to the conclusion should also be shown in the main manuscript not only in the appendix.

- With your approach you assume the same regression parameters for all locations. Could a geographically weighted regression be more useful in case of spatially autocorrelated residuals? Especially, since you expect spatial differences.

**Specific comments**

- line 106: I guess this is known for a much longer time. Maybe this source (doi:10.1111/j.1752-1688.1975.tb00689.x) is more appropriate.

- line 145: remove one "A"

- section 2.1.1: Does every household have a flood insurance?

- Section 3: Is the heating correlated with the age?

- Line 272-275: Is this really the case? The increasing effect on the damage costs are only distributed on more variables in model 2 and 3. I guess, if you would predict damage costs the predictions would not differ that much.

- Line 297: This is not sufficiently supported by your results. You have not investigated the accuracy of the models in terms of predictions yet.

---

## Referee Comment (RC2) · Anonymous Referee #2 · 26 Jun 2020

General comments:

This study uses a dataset of damage costs from building-level insurance claims to build a multivariate cost model for a storm surge event which occurred in Denmark, and finds that adding variables from a variety of sources (emergency response data, building/household data, past flood experience) improves the model's explanatory power over a simple depth-damage approach.

The subject matter is interesting, and the use of a novel set of explanatory variables with insurance data from a storm surge event merits this manuscript's inclusion in the published literature on this topic. However, I find the manuscript to be confusing in

sections from a structural and content perspective. A high-level comment is that depth-damage models must use a "relative" damage degree to be valid (damage relative to total value). Including "building area", "building size", "building type", "quality of heating/cooling system" or some other proxy for "total value" will always improve such a damage model! This is a flaw with the manuscript: the "simplest" model should use both inundation and floor area as explanatory variables if "total insured value" (TIV) or "replacement cost value" (RCV) or some other value cannot be used to make damage a "relative" variable. In this sense, "Model 1" should use both "depth" and "size), or an equivalent model should be added using both "depth" and "value" As mentioned before, all "damage costs" would should ideally be normalized by "insured value" or "replacement cost". This is the biggest flaw in the study and should be addressed before acceptance.

Overall, the paper could be improved by (a) shortening the literature review, (b) clarifying some of the methods, and (c) separating results from the discussion to create a clearer and more distilled message. The readability can be improved substantially with relatively minor edits to the structure and content.

Specific comments:

Section 1 (the introduction) is too long and the literature review should be edited to improve the flow of the paper and properly introduce the subject without too much repetition. The paragraph from lines 37 – 48 would be more effective if it were moved towards the end of the introduction and combined with other sentences describing your study. This is not essential, but papers are usually most readable when the "motivation" and structure of the paper come at the end of the introduction, after existing studies have been discussed. The paragraph from lines 49 to 60 is vague in its discussion of the existing literature (see detailed comments below) and should more explicitly mention the specific contributions from the papers that are cited. In lines 61-64, you essentially repeat the point made from lines 49-60.

[Figure]

Section 2.1 (data) in the Materials and methods section can also be shortened. In Section 2.1.2, it is not clear to me how the simulation approach works, or which method was used for the model development in the paper. Do you run both a bathtub model which uses the "nearest gage"? How does the "nearest gage" depth approach differ from the static 2d model? This section should be edited for clarity and simplified (it sounds like lines 119-123 contain the full approach). Any literature comparing "bathtub" or "static" surge models with hydrodynamic models for modeling storm surge events would be helpful here as well to support the approach.

Section 3, OLS regression results, is too long and dense, and overinterpretation of the results gets in the way of a good discussion. I think this would be better split into a "regression results" and then "discussion" section rather than mixing results with discussion/interpretation. The section from line ∼200 to line 265 is too long and is confusing. It would be better to shorten this section substantially; Table 2 is sufficient to get the main message across. For some of the variables into the model (e.g. building characteristics), there is a nice explanation. For others (e.g. share of retired persons, expenditure on medical consultations) the interpretation of the results is confusing, and these variables may be interrelated.

Technical Corrections:

Line 24 – Just an opinion: "a storm surge" sounds unnatural. It would be better to use "a storm surge event" or some such phrasing. In line 79 "storm surge-induced coastal flooding" is used which also seems like a good phrase to use.

Line 33 – I would say ".. using surveys of affected communities". "Survey methods" is vague.

Line 35-36 – This sentence is a little confusing – the comma is not used correctly here as "insured damage costs being of interest to insurance companies" is not a separate (parenthetical) statement.

Line 41 – I would say "many" uncertainties instead of "a lot of".

Line 50 – "while at the same time highlighting the need for proper data" – I think you can cut this phrase, unless there is a more specific point you'd like to make.

Line 52 – "improve the performance when transferred to other settings both temporally and spatially." This is again a vague phrasing – I'm not sure what is meant here, and likewise Lines 53-54 "However, having used several damage-function models drawn from the literature, they also emphasize that the overall structure of the model is a controlling factor in its performance." This is too general.

Line 89 – "levels" should probably be "spatial level", or maybe "spatial scale" (whichever you use, keep it consistent with the words used in Table 1)

Table 1 – what is "Tokes Data (?)". Also, why use three "distance to waterbody" measurements in the model? What is the difference between distance to lake, coast, and harbor? If this is storm surge flooding, distance to lake should have no impact at all. And distance to coast and distance to harbor are confusing because they sound the same.

Line 130-131 – "the model does not account for the natural infiltration and filling of drainage systems" – what is meant here? If there are specific drainage systems in place in these municipalities to mitigate storm surge hazard, they should be mentioned here (separate from the physical barriers used in the water depth model).

Line 133 – "Building" rather than "buildling" characteristics (header of 2.1.3 misspelled)

Lines 163-165 – Section 2.1.7 can be removed after moving Lines 163-165 to section 2.1.1. I would argue that this section is redundant after making that restructuring move.

Lines 221-223 – I do not understand this sentence: "The hypothesis is that the vulnerability of a local community could influence the municipality's overall efforts to reduce the expected damage costs so that municipalities that are more vulnerable have less economic and social capital with which to respond to extreme events such as storm

surges." It is a run on sentence and seems to make a number of vague statements and suggestions.

Line 227 – what is a % increase in inundation depth? Relative to maximum inundation depth in the entire dataset? In the municipality? This should probably be expressed in a % increase in damage per meter of increased inundation depth.

Line 231 – what is the "size of the building"? The size in square meters of floor space? Building value? Number of stories?